# Kindergarten Teachers’ Perspectives on Young Children’s Bullying Roles in Relation to Dominance and Peer Relationships: A Short-Term Longitudinal Approach in South Korea

**DOI:** 10.3390/ijerph17051734

**Published:** 2020-03-06

**Authors:** Seung-ha Lee

**Affiliations:** Department of Early Childhood Education, Chung-Ang University, Seoul 06974, Korea; seungha94@gmail.com

**Keywords:** bullying, victimization, dominance, resource control strategies, young children

## Abstract

There are several studies on young children’s bullying roles in relation to dominance or peer relationships. Although those are closely related, few studies examined this from longitudinal view and the influence of bullying role change on dominance and peer relationships. This study aimed to examine (1) the relationship between bullying roles and dominance, (2) the relationship between bullying roles and peer relationships, (3) the percentage of bullying role change over time, and (4) the changes in bullying roles in relation to changes in dominance and peer relationships. Sixty-three South Korean kindergarten teachers completed questionnaires regarding bullying roles, dominance, and peer relationships about 1312 children aged 3–5. The data were collected in mid-October 2017 and January 2018. The results showed that bullies had the highest dominance. No-role children had the most positive peer relationships, followed by bullies. About 10% of all sampled children remained involved in bullying over time. Their role changes related to changes in dominance rather than to changes in peer relationships. The findings imply that dominance should be considered to prevent young children’s bullying, in which peer relationships are interrelated. Intervention should be implemented as soon as possible to stop repeated victimization or bullying in early childhood.

## 1. Introduction

Bullying is a pervasive phenomenon across all age ranges. Research on bullying has predominantly focused on middle childhood or adolescence, but some studies have reported that bullying clearly occurs among young children in preschools or kindergartens [1,2,3,4,5]. Detecting bullying among young children is necessary to prevent social and behavioral developmental problems. Bullies in kindergarten appear low on cooperativeness and prosocial behavior [6], and bullying is associated with low social preferences among young children [7]. Longitudinal studies have shown the link between victimization experiences and adjustment problems more clearly. Children who were relationally victimized in kindergarten showed high levels of loneliness and school avoidance, and low levels of school liking [8]. Moreover, young children’s victimization or bullying experiences resulted in internalizing and externalizing problems later on. Five-year-old pure victims later displayed more internalizing problems and unhappiness at school when aged 7. Bully/victims also showed more internalizing and externalizing problems than control groups or pure victims upon reaching the age of seven [9].

Bullying has been defined as an intentional aggressive behavior repeatedly inflicted over time against a targeted individual who finds self-defense difficult [10]. Therefore, the power imbalance is an important element in defining aggressive behavior as bullying. Conflicts between peers who have similar levels of power (the power can be physical or social) are not considered bullying [11].

Broadly, there are two perspectives to explain bullying. From the perspective of the Social Information Processing (SIP) model, bullies are viewed as socially unskillful and inept. This fits the traditional view of bullying; bullies are unlikely to understand others’ views or thoughts [12], and tend to lack affective empathy [13]. From this perspective, bullies’ aggressive behaviors result from their misinterpretations of social cues, which leads them to behave in socially inappropriate ways. In contrast, Hawley’s Resource Control Theory (RCT) explains that bullies can be socially competent. According to this perspective, some bullies can be competent in interpreting others’ intentions, and use aggressive behavior to achieve their goals—such as obtaining their high peer status or acquiring valuable resources [13,14]. Consistent with RCT, some other studies have also shown the high ability in cognition among bullies. Bullies aged 7–10 show higher ability in social cognition and emotional understanding [15]. Moreover, there are positive correlations between bullying behaviors and cognitive empathy among 8–10-year-old children; in contrast, no relationship between bullying behaviors and affective empathy within the same sample has been found [13]. In addition, even young bullies (aged 6) show the higher theory of mind scores that non-bullies [16].

### 1.1. Bullying, Dominance, and Peer Relationships

Young children tend to use an overt form of aggression more than a covert form of aggression. If they use relational aggression, it tends to be direct relational form, such as ‘rejecting a peer by telling the child directly ‘you are not my friend’ ‘or’ placing one’s hand on a chair, thus, a child was not allowed to sit at the table’ [17]. Moreover, this form of aggression is related to gender; girls are more likely to be relationally aggressive than boys [18].

Dominance can be conceptualized as a power, resulting in a peer’s submission [19,20]. Therefore, bullying is essentially related to dominance, as it is a strategic behavior used to obtain powerful status among peers [21]. This is also observed in bullying or aggressive behavior in early childhood [22,23]. Socially dominant girls tend to be more relationally aggressive than their peers in early childhood [23].

Dominant individuals are more likely to obtain physical resources or social reputation, such as attractive toys, peers’ attention, or popularity. According to RCT, both prosocial and aggressive behaviors can be used to obtain valuable and limited resources. Coercive strategies comprise aggressive tactics to control resources, such as monopolizing, threatening, and assaulting. In contrast, prosocial strategies include prosocial behaviors for controlling resources, such as reciprocation, alliance formation, or cooperation [14,24]. Dominant children tend to use both coercive and prosocial strategies (i.e., bistrategics) rather than one type of strategy [25,26]. Young children can use either coercive strategies to obtain resources, such as taking away toys or threatening, or prosocial strategies, such as negotiating, exchanging, or promising friendship [14,27].

Dominance can differ from bullying roles. Pro-bullying roles (ringleader bullies, assistants, and bully/victims) mostly tend to be bistrategic resource controllers among 4th, 5th, and 6th grade children [19]. Ringleader bullies, in particular, have the highest desired dominance (i.e., how much they want to be dominant) and the highest resource control (that is, acquired dominance) levels. Meanwhile, victims show the lowest resource control. Moreover, bully/victims are close to, but slightly lower than ringleader bullies in terms of their resource control and dominance [19].

Distinguishing bully/victims from bullies is important. Bully/victims tend to use high levels of both proactive and reactive aggression, just as bullies do [28,29]. However, their behaviors tend to be more emotionally dysregulated and less strategic than those of bullies [28]. This may lead bully/victims to be less socially dominant than bullies. Olthof et al. [19] emphasized the importance of the relationships between bullying roles, social dominance, and coercive and prosocial strategies. However, very few studies have examined the direct relationship between these factors [30].

Research findings regarding the link between peer relationships and bullying roles have been inconsistent. Peer relationships have often been examined through likeability or popularity using sociometric methods. Popularity and likeability are often used interchangeably, although they are not exactly the same; popularity sometimes refers to visibility, dominance, and power among peers (i.e., perceived popularity), whereas, likeability represents the degree of being liked by peers (i.e., peer acceptance, peer rejection) [31,32].

Generally, bullies experience high levels of rejection and low levels of acceptance, whereas, victims are neither particularly disliked nor liked by their peers in early childhood, based on peer reports [4,7,33]. In contrast, other studies have shown that 5–7-year-old bullies have higher social status and larger friendship networks than bullies/victims or victims [6,34]. Similarly, aggressive preschoolers tend to be popular or have the similar social status to children who are not involved in bullying, whereas, the majority of bully/victims are likely to have rejected or controversial peer status [35]. Moreover, there does not appear to be a negative relationship between bullying and acceptance [36].

The inconsistent findings on the relationship between peer acceptance and bullying roles can be partly explained in terms of dominance. Popularity is deeply associated with bullying when both affiliative and aggressive strategies are used together [37]. Bullies in mid-childhood are perceived as popular and having power over others [38], while victims in mid- childhood have low resource control abilities and low perceived popularity [19]. Victimization may lead to loss of friendships or, at least, playmate relationships, because children might think that being friends with victims is unattractive [6].

### 1.2. Dominance and Peer Status in Relation to Changes in Bullying Roles

Many studies have indicated that bully roles are more stable than victim roles, and this tendency is more pronounced in early childhood [36,39,40]. Generally, aggressor roles tend to be quite stable for two months [41], four months [4], or 18 months [42]. Victim roles, meanwhile, are more changeable in early childhood [4,41,43,44], which is not the case among older children or adolescents. This may reflect that preschool bullies may be more likely than older children to be aggressive toward a large number of children rather than specifically targeted children [4,45].

Some children are repeatedly victimized for long periods of time; others are not. Generally, studies have reported that bullies or bully/victims are more likely than victims to maintain their roles over time. According to peer reports, among 7–8-year-old children, 40% of bullies, 17% of victims, and 55% of bully/victims retained the same roles between the two measured time points over a 1-year period [46]. Among 4–6-year-old children, 60% of aggressors and 13% of victims remained in their roles over a 4-month period [4]. Meanwhile, Schäfer et al. [47] found that 32% of bullies, 20% of victims, and 12% of bully/victims in early primary school retained the same roles over a 6-year period. Wolke et al. [48] conducted a longitudinal study to examine the stability of victimization among children aged 6–9 over a 2–4-year period. They showed that the probability of remaining in a victim role was almost two times that of becoming a new victim; 38% of victims at time 1 remained victims at time 2, whereas, 20% of non-victims at time 1 became new victims at time 2.

Peer hierarchical structure and dominance can explain the stability/changes in bullying or victimization roles [46,47,48]. Children from classes with highly hierarchical structures were more likely to remain as victims of relational aggression [48]. Strong peer hierarchies can make it difficult for children to escape victimization; victims need power to stand up to bullies to escape, which may be difficult for them because of their weak peer status and the rigidity of the hierarchy in their classes.

The causality between bullying and social dominance is not clear. There are inconsistent findings.

Some studies have indicated that bullying is not necessarily used to obtain dominance. Bullies increase their bullying to achieve dominant status, especially when they enter new schools [49]. Reijntjes et al. [30] examined the causality between bullying and dominance in their three-year longitudinal study. They found that children who maintained a high level of bullying showed consistently high levels of resource control, but the reverse was not true. That is, bullying was being used to obtain dominance; however, dominant children did not necessarily bully others. However, another study showed that bullying behaviors did not increase when entering a new group; that is, bullying did not seem to be used to obtain dominance [39]. Changes in bullying roles may interplay with changes in dominance and peer relationships. Sentse et al. [32] examined the longitudinal interplay between bullying, victimization, likeability, and perceived popularity among children in grades 3–6 over a 1-year period. They found that victimization and peer acceptance have bidirectional influences on each other—victimization negatively affects peer acceptance and vice versa. Similarly, victimization and peer rejection have a positive association and bidirectional influences among both girls and boys in primary school pupils [32]. Moreover, perceived popularity negatively influences victimization for boys, and positively affects bullying for girls [32]. Other studies indicate that bullies may have high perceived popularity [13], while having low social preference [7,13]. Furthermore, the relation between peer acceptance/rejection and bullying may differ by children’s sex; Children’s likeability toward a bully depends on whether the bully is of the same sex. Girls are more likely to reject bullying girls than bullying boys, which may reflect that bullying in boys may be seen by girls as a boyish feature rather than a bad behavior [50]. Similarly, girls do not dislike boys who bully boys in contrast to girls who bully boys, which shows a low level of acceptance by both sexes [36].

### 1.3. Bullying and School System in South Korea

Bullying is an English term, and thus, the phenomenon corresponding to bullying may differ across cultures. There are several Korean terms which correspond to the term bullying; *hakkyo-pokryuk* (school violence), *gipdan-goerophim* (collective teasing or collective bullying), *gipdan-ttadolim* (collective isolation), and *wang-ta* (social exclusion or an excluded/victimized person). These terms are used interchangeably, although each term includes slightly different types of aggressive behavior [51]. Recently, the term *hakkyo-pokryuk* has been widely used in public as it includes a wide range of behaviors that occur among school pupils (i.e., physical and verbal aggression, extortion, social exclusion, cyber aggression, and sexual harassment).

The cultural differences in bullying are explained in terms of Western and Eastern or collectivistic and individualistic dimensions [52]. South Korea is a collectivistic culture, in which interdependency and harmony among in-group members are more prioritized than independency and individual achievement [53]. The characteristics of *hakkyo-pokryuk* in South Korea may reflect more the collectivistic nature on aggressive behavior than in Western societies. For example, collectivistic cultures, such as South Korea or Japan show a higher proportion of aggressors to victims than those of Western countries [54,55]. Moreover, in South Korea, there are terms that indicate socially excluded or victimized persons, as well as several slang terms, depending on the severity of exclusion among pupils [51]. For example, *wang-ta* means complete exclusion or excluded person, which is a slang term popularized among school pupils in the late 1990s. Labeling a victim with this term can more seriously stigmatize the victim.

The Korean education structure is organized into six years of elementary school, three years of middle school, and three years of high school. In early childhood, 3–5-year-old children can be registered for either kindergarten or a daycare center. Indeed, most young children (90%) aged 3–5 in South Korea are registered either in kindergartens or daycare centers [56]. Therefore, young children in South Korea experience social relationships in public educational settings. International studies have shown that bullying clearly exists among young children [1,3,4]. However, most studies about bullying in South Korea have predominantly focused on school-aged pupils. Only limited studies have been conducted on the bullying of young children in South Korea. For instance, Kwak and Kim [57] investigated kindergarten and daycare teachers’ perceptions of bullying. They showed that 63% of teachers witnessed young children’s bullying and reported the necessity of an intervention program.

### 1.4. Needs for the Study

Bullies tend to have higher social dominance than non-bullies, and bullying roles are related to peer status. This raises questions regarding whether remaining in the same bullying roles relates to the maintenance of dominance or peer status. Role changes is an important issue in defining bullying, because repetition of aggressive behaviors is a main characteristic of bullying. Previous studies examining the stability of bullying in early childhood have provided useful information regarding the consistency of each bullying role. However, they have not tracked changes in bullying roles. For example, some bullies may remain bullies, but others may move to different roles, such as bully/victims or victims. Moreover, if a child escapes from victimization or becomes newly involved in bullying, their peer relationships or dominance also presumably changes.

While the links between bullying and peer relationships and between peer relationships and dominance have been extensively researched, almost no studies have examined the relationships among all of these factors (bullying, dominance, and peer relationships) together, especially in early childhood. Some previous studies about this topic have been conducted, but specifically based on Western cultures (e.g., in the U.S.). Furthermore, research regarding changes in bullying roles in relation to dominance and peer relationships remains scarce.

### 1.5. Aims of the Study

This study aims to examine:(1)The relationship between bullying roles and dominance;(2)The relationship between bullying roles and peer relationships;(3)The percentage of bullying role change; and(4)Changes in bullying roles in relation to changes in dominance and peer relationships.

## 2. Method

### 2.1. Participants

Sixty-two teachers (Mean = 35.0 years, SD = 8.01), aged from 24 to 53, from 29 public kindergartens in city A, South Korea participated in this study. The number of classes participated in this study differed by kindergartens as each kindergarten had a different number of classes. The range of the number of classes for one kindergarten which participated in this study was from one to eight; averagely, 2.13 classes per kindergarten participated in this study. Each teacher managed their own classes (62 classes total), with a combined total of 1324 children aged 3–5. The teachers were all women. The researcher did not intend to examine participants of single sex, but the predominance of women likely results from the fact that most kindergarten teachers in South Korea are women. The teachers were recruited through a notice board at the Early Childhood Education and Promotion Center, which is directly affiliated with the Office of Education in South Korea. Teachers who voluntarily agreed to participate were contacted by phone, and the aim of the research was explained to them individually. If they again agreed to participate, the questionnaire was sent to them. Teachers’ written consent was also obtained. The teachers who agreed to participate in this study completed the questionnaire twice, with two months between the two time points. The number of measuring was decided by considering the methodologies of previous studies and teachers’ daily workload in the kindergarten. Some previous studies [4,41] measured children’s bullying or peer relationships twice to investigate changes of those over two- or four-month period. Moreover, the kindergarten teachers have a lot of work to manage their classes, thus, it was difficult for them to measure children more than twice within one semester. Thus, measuring twice over two months was regarded as appropriate. The participants received a 100,000 Korean won (corresponding to 80 U.S. dollars) gift certificate for their participation.

### 2.2. Procedure

All of the measurements were originally developed in English, so they were translated into Korean by the author. An independent bilingual researcher back-translated from Korean to English. Next, the original questionnaires developed in English were compared to the back-translated questionnaires in English. As discrepancies between the original questionnaire and the back-translated questionnaire were rarely found, the Korean translation was judged to be appropriate.

To confirm the appropriateness of the measurements, 13 kindergarten teachers were asked to complete the questionnaire in a pilot study. They understood all items in the questionnaire, and had no difficulty in completing it. Teachers were knowledgeable about the peer relationships among the children in their class, as they had spent seven months with them. Furthermore, kindergarten teachers in South Korea always stay with their children during the day. Previous studies have shown that teacher reports are a valid method for measuring young children’s bullying, since preschool teachers spend a great deal of time with their students [7,33,58]. It has been shown that teacher reports reliably measure young children’s bullying roles [7,32] social dominance across a 4-month period [59], as well as children’s peer acceptance and rejection and prosocial and coercive strategies [60].

One teacher in the pilot study suggested that the like most (LM) and like least (LL) measures would be more accurate if teachers asked the children about it directly. Therefore, teachers in the main study were allowed to ask their children directly about peer relationships where possible. Some teachers did not agree with this approach and preferred to respond to the question themselves. Furthermore, some teachers regularly checked with the children regarding their friendships by observing them. Therefore, in the main study, teachers rated each child’s LM and LL by observing the child’s peer relationships in during kindergarten. Moreover, they were able to ask children in individual interviews if they were unsure of the appropriate response.

This study relied on teachers’ reports for collecting data, which was unavoidable, due to the decision of the Institutional Review Board (IRB). For the reliability and validity of the data, the influence of classroom variables, such as teachers’ carrier, sex ratio (the ratio of boys in relation to girls in one class), and class size (number of children in a class), as well as the method for investigating –LM and LL (i.e., observation or asking directly), were included in the analysis.

Teachers completed the questionnaires for the 1324 children (Mean = 4.5 years; SD = 0.687) in their classes. All variables—bullying roles, peer relationships (LM and LL peer acceptance among same sex children, peer acceptance among opposite sex children), and dominance (social dominance, prosocial strategies, coercive strategies)—were measured twice, once in October 2017 (Time 1) and once in January 2018 (Time 2), with an average of 9.5 weeks between the two time points in each kindergarten. The overall data collection was completed within 12 weeks (from October 2017 to January 2018), including the period of distributing and collecting the questionnaires.

Previous studies used diverse time periods for examining the stability of children’s behavior or peer relationships; one week [61], two months [41], four months [4], or eighteen months [42]. The time period of this study was restricted by the semester system in kindergartens: The first semester from March to July, and the second semester from September to December. If holidays or vacations occur between the two time points, it might influence the results of bullying or social dynamics; therefore, the two measurements should be taken within one semester. For this reason, the time points of the measurements of this study were decided from October 2017 to January 2018; the questionnaire distributed for T2 was returned back after Christmas. In order to prevent the effect of teachers’ first measurement to the second, at Time 2, teachers were asked to measure children’s behavior and peer relationships during the “recent” two months after the first measurement.

### 2.3. Instruments

#### 2.3.1. Bullying Roles

Teachers responded to questions regarding the bullying behaviors and victimization of each child in their class, indicating how frequently children engaged in (aggressor) and were the recipients of (victim) these behaviors. Regarding physical and verbal aggression, three items were adapted from Ostrov, Pilat, and Crick [60], and Crick and Grotpeter [62] (e.g., “The child hits/pushes other children”, “The child takes away other children’s toys”, and “The child calls other children mean names). Five items for relational aggression were adapted from Crick, Casas, and Ku [63], Crick, Casas, and Mosher [64], and Nelson et al. [35] (e.g., “The child does not allow other children to play”, “This child tells other children not to play with a particular child”, “This child spreads rumors about other children”, “This child ignores other children”, and “This child says ‘you are not my friend’ if other children do not comply with his/her request”). Participants selected responses on a 5-point Likert scale from 1 (completely disagree) to 5 (completely agree). Victimization was measured using items comparable to those used to measure the aggression—3 items for physical and verbal victimization (e.g., “Other children hit/push this child”, etc.) and five items for relational victimization (e.g., “Other children do not allow this child in play”, etc.). The bullying and victimization measures showed high reliabilities; the Cronbach’s alphas were 0.933 (T1) and 0.943 (T2) for the bullying items, and 0.930 (T1) and 0.934 (T2) for the victimization items. Children were assigned to the bullying role based on their bullying and victimization scores. Previous studies used various standards, from 0.1 SD to 1 SD [4,32], for this categorization. This study used 1 SD as a standard for categorizing children into the bullying role, because it may provide a clearer distinction between the bullying behaviors and victimization experiences than 0.1 SD. The mean scores of bullying and victimization were standardized. Children that scored above 1 SD in bullying and below 1 SD in victimization (Z bullying score >1.0 and Z victimization score <1.0) were categorized as bullies; children that scored below 1 SD in bullying and above 1 SD in victimization were categorized as victims; and children that scored above 1 SD in both bullying and victimization were categorized as bully/victims. The remainder of the children were categorized into the no-role group. These categorizations were applied both for Time 1 and Time 2. This study applied four roles, including non-involved children (bully, victim, bully/victim, no roles), although there were six participant roles (ringleader, assistant, reinforcer, victim, defender, and outsider) among traditional studies [4,65]. The reason for applying four roles in this study was that there were 16 possible cases of role changes; one role may have three cases of role change and one maintenance case (e.g., bully → bully, bully → victim, bully → bully/victim, bully → no role). If six participant roles were used, it would be very complicated to track all of the cases of role change. Additionally, previous studies have shown valid results using less than six roles for young children’s bullying [6,7].

#### 2.3.2. Peer Relationships: Like Most, Like Least, Peer Acceptance among Same Sex, and Peer Acceptance among Opposite Sex 

##### Peer Nomination

A peer nomination measure was used to examine like most and like least. Like most (LM) was measured by asking “Whom does this child like most?”, and like least (LL) was measured by asking “Whom does this child like least”? For each question, participants were asked to make up to three nominations [4,66], but were allowed to make more than three if needed [67]. The number of LM nominations each child received was summed and standardized across each class, and the LL scores were computed in the same way. The standardized scores were used in all of the analyses. If available, teachers asked children directly about LM and LL. Almost half of the teachers observed children and recorded children’s LM and LL—33 teachers at T1 (53.2%) and 30 teachers at T2 (48.4%); 12 (19.3%) teachers at T1 and 17 (27.4%) teachers at T2 asked children directly about LM and LL; and 16 (25.8%) teachers at T1 and 13 (20.7%) teachers at T2 used both observation and asking children directly, and one teacher did not reply to the methodology.

##### Peer Rating

Peer rating was used to measure a child’s peer acceptance by the same and opposite sex children; the questions from the Preschool Social Behavior Scale-Teacher Form (PSBS-TF) were used [60]. Teachers provided responses on a 5-point scale (from strongly dislike to strongly like) to the following items: “How much is this child liked by same sex peers?” (PASS: Peer acceptance among same sex) and “How much is this child liked by opposite sex peers?” (PAOS: Peer acceptance among opposite sex).Previous studies have shown teachers’ reports of peer nominations for LM and LL and peer ratings to be a reliable measurement method [60,64,68].

#### 2.3.3. Dominance

Dominance was measured using two constructs: Social dominance and resource control strategies.

##### Social Dominance

Social dominance (SoD) in this study referred to already-acquired dominance and was distinguished from the behavior used to obtain dominance [17]. The questionnaire measured social dominance using instruments derived from previous studies [14,23,27,69] (e.g., “This child gets what he/she wants even if others do not”). The measure consisted of six items on a 5-point Likert scale, from 1 (never) to 5 (always). The *Cronbach’s alphas* for this measure were 0.893 (T1) and 0.911 (T2).

##### Resource Control Strategy

The resource control strategy items in the questionnaire were derived from Hawley’s work [27]. They included six prosocial strategies (PS) of control items (e.g., “S/he is someone who influences others by doing something in return”) and six coercive strategies (CS) of control items (e.g., “S/he pushes other to do what s/he wants”), all measured on 7-point scales from 1 (never) to 7 (always). The items showed high reliability: The Cronbach’s alphas were 0.929 (T1) and 0.943 (T2) for the prosocial strategy items, and 0.949 (T1) and 0.951 (T2) for the coercive strategy items.

All of the questionnaires regarding the 1324 children were collected for both times 1 and 2. Twelve incomplete questionnaires—due to children’s absence or departures from the surveyed kindergartens—were exempted from the analysis. Moreover, the questionnaires missing several variables within one dependent variable (e.g., several items of the prosocial strategy were missed) were excluded from the analysis. The percentage of missing values was 0.1~0.4. Finally, 1312 children’s data measured at both Time 1 and Time 2 were used for the analysis.

### 2.4. Ethical Issues and Consent

This study obtained IRB (Institutional Review Board) approval from the researcher’s university (IRB number: YU201707001003). The IRB did not allow the researcher to approach children to ask them directly about their peer relationships and bullying experiences; such questions, when posed by an unfamiliar adult, could have been stressful or uncomfortable for them. The IRB, therefore, required that the researcher collect data through kindergarten teachers.

To increase (maximize) the reliability of the collected data, this study tried to ensure that the measurers’ (i.e., the teachers’) backgrounds were consistent. There are two types of kindergarten in South Korea: Private and public. To become private kindergarten teachers, individuals must obtain kindergarten teacher licenses from college or university early childhood education departments. To become teachers in public kindergartens established at the national level, licensed kindergarten teachers must pass the national teacher employment examination. Public kindergarten teachers generally have similar academic backgrounds and working environments, and they all apply the national educational curriculum—the most representative kindergarten curriculum in South Korea. Therefore, this study collected data from public kindergarten teachers in South Korea. Moreover, observing and recording the behavior of all the children in their classes on a regular basis is also a very general duty for both private and public kindergarten teachers. Therefore, the teachers in this research were keen to observe their children in the class.

On the questionnaire itself, the teachers encoded all children’s names as random numbers, so the researcher did not know the children’s names or their exact ages. This study required only the mean age and sex of the children in the classes. In South Korea, most classes in kindergarten are organized by children’s ages (such as 3, 4, and 5). According to the IRB’s decision, information beyond age and sex was neither needed nor available.

### 2.5. Statistical Analysis

SPSS 25 version was used to analyze the data. Before examining the four aims of this study, it was necessary to see the overall patterns of the data. Moreover, it is required to understand whether class-related variables were involved in the levels of bullying, victimization, SoD, PS, and CS as bullying and victimization may be related to classroom dynamics, and children are nested within classrooms. First, means and standard deviations were calculated for raw scores of bullying, victimizations, SoD, PS, and CS. Second, pearson’s correlations among all variables in this study were conducted: Bullying, victimization, SoD, PS, CS, LM, LL, PASS, PAOS, and class-related variables. Class-related variables indicate the variables at the classroom levels: Teachers’ carrier (year of teaching experience), sex ratio (if the ratio of boys in a class is higher than 0.5, it was coded as 1, and the remainder was coded as 0), class age (3-, 4-, and 5-year-old children), and class size (number of children in a class). Third, multiple regression was conducted to examine whether class-related variables were associated with the scores of bullying, victimizations, SoD, PS, and CS. In each multiple regression analysis, class-related variables, such as teachers’ carrier, sex ratio, class age, and class size were the predictors, and bullying, victimization, SoD, PS, and CS were the outcomes. Fourth, after categorizing children to bullying roles, chi-squared tests were applied to investigate the differences in distribution of bullying roles by class-related variables.

To examine aims 1 and 2, a one-way MANOVAs (Multivariate Analysis of Variance) were conducted. A MANOVA for SoD, PS, and CS as the dependent variables (DV) and bullying roles as the independent variable (IV) was conducted. Similarly, for peer relationships variables—LM, LL, PASS, and PAOS—a MANOVA was conducted by bullying roles. Each MANOVA showed significant mean differences across all DVs by bullying roles. Further analyses using univariate tests showed significant differences by bullying roles on SoD, PS, CS, LM, LL, PASS, and PAOS; therefore, post-hoc (*Scheffe*) tests were conducted for all possible pairs of means of each DV across bullying roles.

No statistical test was used for aim 3, since it concerned the percentage of bullying role change. For aim 4, for each case of role change, paired *t*-tests was conducted to examining the mean difference between T1 and T2 for each dependent variable—SoD, PS, CS, LM, LL, PASS and PAOS. Additionally, hierarchical multiple regression was conducted to examine whether the relationships between the change of roles and the change of dominance and peer relationships were affected by class variables (“teachers’ carrier”, “sex ratio”, “class size”, and “class age”) and the method for investigating LM and LL (i.e., asking about LM and LL directly or not).

To perform the hierarchical regression analysis, children who maintain their role over time (e.g., bully → bully) were coded as 0, and children who change their role (e.g., bully → other roles) were coded as 1. Therefore, four groups were created (Group 1: Children who maintain a bully role vs. children who change into another role; Group 2: Children who maintain a victim role vs. children who change into another role; Group3: Children who maintain a bully/victim role vs. children who change into another role; and Group 4: Children who maintain a no-role position vs. children who change into another role). The difference of each DV (SoD, PS, CS, LM, and LL) between T1 and T2 was calculated (e.g., T1 SoD minus T2 SoD: T1 SoD –T2 SoD). Then, hierarchical regressions were conducted; the difference of each DV between two time points was the outcome, and each group (Groups 1, 2, 3, and 4) was entered as a predictor in model 1, and class variables (“teachers’ carrier”, “sex ratio”, “class size”, and “class age”) and the method for investigating LM and LL were additionally entered as predictors in model 2.

## 3. Results

Means and standard deviations of bullying, victimization, SoD, PS and CS are shown in Table 1.

The correlations among variables are shown in Table 2. Bullying and victimization were moderately or highly related. Moreover, they were positively correlated to SoD, PS, CS, and LM, and negatively correlated to LL, PASS, and PAOS. Sex ratio was positively correlated to bullying, victimization, SoD, PS, and CS, whereas, class size and class age were often negatively related to them.

Multiple regression showed that there were no significant effects of class variables on the means of bullying, victimization, SoD, PS, and CS.

Following the rules of role distinction explained above, 25–26% of the measured children were involved in bullying over time. Table 3 shows the number of children in each role at both T1 and T2. About 10% of children were categorized as bullies. The number of victims increased at T2, whereas, bully/victims decreased at T2.

Additionally, the results of the chi-squared tests showed that none of the class variables was significantly related to the distribution of bullying role.

### 3.1. Relationship between Bullying Roles and Social Dominance and Prosocial and Coercive Strategies

There were significant correlations among the three dependent variables in dominance -SoD, PS, and CS; The correlation coefficients (Pearson’s *r*) among them were from 0.44 to 0.60 at T1, and from 0.51 to 0.56 at T2. A MANOVA showed a significant effect of bullying roles on SoD, PS and CS; *Wilks’s lambda* = 0.654, *F* (9, 3129.937) = 66.398, *p* = 0.000, *partial*
*η*^2^ = 0.13 at T1, and *Wilks’s lambda* = 0.678, *F* (9, 3151.840) = 60.620, *p* = 0.000, *partial*
*η*^2^ = 0.12 at T2. Follow-up univariate tests showed significant mean differences in SoD, CS, and PS across bullying roles.

Bullies showed the highest means in SoD, PS, and CS, followed by bully/victims, victims, and no-role children. Moreover, victims and no-role children generally showed similar means, and only some of the means were significantly different. Table 4 shows the results of the univariate and post-hoc (*Scheffe*) tests.

Bullies and bully/victims were not different in PS at T1, CS at T1 and T2, whereas, bullies showed a higher SoD score than bully/victims at T1 and T2. No significant differences were found in SoD between victims and no-role children.

### 3.2. Relationship Between Bullying Roles and Peer Relationships

There were significant correlations among the four dependent variables in peer relationships—LM, LL, PASS, PAOS; The correlation coefficients (Pearson’s *r*) among them were from −39 to 0.49 at T1, and from −0.37 to 0.62 at T2. A MANOVA showed a significant effect for peer relationships; *Wilks’s lambda* = 0.720, *F* (12, 3355.104) = 36.985, *p* = 0.000, *partial*
*η*^2^ = 0.10 at T1, and *Wilks’s lambda* = 0.725, *F* (12, 3344.521) = 36.055, *p* = 0.000, *partial*
*η*^2^ = 0.10 at T2. A follow-up univariate tests showed significant mean differences in LM, LL, PASS, and PAOS across bullying roles. The results of the univariate tests and further post-hoc tests are shown in Table 4.

No-role children and bullies showed higher LM, PASS, and PAOS and lowered LL than victims and bully/victims. Bullies’ LL and PAOS scores were significantly lower than no-role children, whereas, bullies’ LM and PASS scores were not significantly different from those of no-role children.

Victims and bully/victims had low LM, low PASS, low PAOS, and high LL scores; the differences between victims and bully/victims in these scores were not significant, except for one case; the difference between victims and bully/victims was significant on LL at T2.

### 3.3. Percentage of Bullying role Change

Table 5 indicates the percentages of change in each bullying role. Forty-two percent of children in the bully role at T1 maintained their roles at T2 (*n* = 54) (bully–bully), 39% of victims at T1 maintained their roles at T2 (*n* = 40) (victim–victim), and 29% of bully/victims at T1 maintained their roles at T2 (*n* = 33) (bully/victim–bully/victim). Thus, 36.9% (127 children) of the children involved in bullying at T1 had the same roles at T2.

Eighty-eight percent of children with no roles at T1 had the same role at T2 (*n* = 852) (no role–no role), which was 65% (852 children) of 1312 children. Conversely, 35.1% of children measured in this study did have a bullying-related role either at T1 or T2.

### 3.4. Changes in Bullying Roles in Relation to Changes in Dominance and Peer Relationships

The analysis showed some significant differences in SoD, PS, and CS based on children’s role changes. Table 5 presents the means and standard deviations for SoD, PS, and CS based on role changes over time. Children who maintained their roles as bullies showed no significant differences in their SoD, PS, and CS scores between T1 and T2. However, the scores of children who lost their bully roles decreased for all three variables. Bully–no role (i.e., children whose roles changed from bully to no role) decreased in SoD (*t* (*47*) = 4.82, *p* < 0.001), PS (*t* (*47*) = 3.52, *p* < 0.001), and CS (*t* (*48*) = 5.12, *p* < 0.001). Bully–bully/victim decreased in CS (*t* (*18*) = 2.99, *p* < 0.01).

Victim–bully/victim increased significantly in SD (*t* (*11*) = −2.82, *p* < 0.05) and CS (*t* (*11*) = −2.82, *p* < 0.05). Victim–no role decreased in SoD (*t* (44) = 3.06, *p* < 0.01) and CS (*t* (*44*) = 2.70, *p* < 0.05).

Children who moved from bully/victim to other roles showed decreases in SoD, PS, and CS. Bully/victim–bully decreased in both PS (*t* (*29*) = 3.79, *p* < 0.01) and CS (*t* (*28*) = 3.22, *p* < 0.001). Bully/victim–bully/victim decreased in SoD (*t* (*32*) = 2.20, *p* < 0.05), while bully–no role decreased in CS (*t* (*27*) = 5.21, *p* < 0.01).

Children who were newly engaged in bullying from no role at T1 showed increases in SD, PS, and CS. No role–bully increased in SoD (*t* (*40*) = −2.12, *p* < 0.05) and CS (*t* (*40*) = −3.52, *p* < 0.01), no role–bully/victim increased in CS (*t* (*19*) = −2.25, *p* < 0.05), and no role–no role children decreased in PS (*t* (*849*) = 4.60, *p* < 0.001.

Only three differences in peer relationships were significant in relation to role changes. Bully/victim–bully/victim decreased in LM scores (M = −0.52, SD = 0.685 at T1, and M = −79, SD = 0.698 at T2; *t* (*32*) = 2.502, *p* < 0.05). No role–victim increased in PASS score (M = 2.93, SD = 0.720 at T1, and M = 3.18, SD = 0.834 at T2; *t* (*44*) = −2.693, *p* < 0.05). No role–bully/victim increased in LL scores (M = −0.05, SD = 0.749 at T1, and M = 0.54, SD = 1.230 at T2; *t* (*16*) = −2.322, *p* < 0.05).

Hierarchical regression models for examining the relationship between role change and changes in SoD, PS, CS, LM, LL, PASS, and PAOS, including class related variables, showed some significant cases. Table 6 shows the results of group 1 (bully group). Children whose role changed from bully to other roles positively predicted SoD, PS, and CS differences over time. That is, children who lost their bully role showed large decreases in SoD, PS, and CS at T2, whereas, children who maintained their bully role tended to have little differences in SoD, PS, and CS at T2. When class variables were added, only class size was significant for predicting PS differences in model 2. Class variables were not significant predictors for SoD or CS differences in Group 1. The hierarchical regression models for predicting LM and LL differences were not significant.

In group 2 (victim group), no hierarchical models were significant. In group 3 (bully/victim group), one hierarchical model was significant, as shown in Table 7. Only role change was a significant predictor for the difference in CS; Children who lost their bully/victim role showed larger decreases in CS at T2.

Hierarchical regressions were significant in group 4 (no role) for SoD, PS, CS, and LL differences, but in the final model, only 0.03% of the difference for LL was explained by the variables. Table 8 shows the results of group 4. Children who were newly involved in bullying roles (role change) showed increases in SoD, PS, CS, and LL scores at T2. When class variables were added to model 2, teachers’ carrier positively predicted decreases in PS and CS. Moreover, the more boys in a class, the higher the SoD difference from T1 to T2.

## 4. Discussion

This study examined young children’s bullying roles in relation to dominance and peer relationships. The findings are discussed in relation to the aims of this study.

The first aim of this study was to examine the relationship between dominance and bullying roles. In this study, SoD, PS, and CS were deeply related to bullying roles. These findings are consistent with previous studies [14,19,27]. Bullies showed the highest SoD, PS, and CS. Interestingly, bullies and bully/victims frequently used both PS and CS at similar levels (although bullies were slightly higher in PS than bully/victims), but bullies showed higher SoD than bully/victims. Bullies seem to use the two strategies (PS and CS) more successfully than bully/victims. This may have resulted from bullies’ frequent use of PS; bullies used PS more frequently than all other roles. PS seems to be an effective means of obtaining SoD when used with CS.

The second aim of this study was to investigate the relationship between bullying roles and peer relationships. The peer relationships of children who were involved in bullying differed by their bullying roles. Children not involved in bullying had the most positive peer relationships. Bullies’ peer relationships were more similar to no-role children’s than victims’ or bully/victims’. Victims and bully/victims had negative peer relationships; actually, victims often had lower peer acceptance and higher peer rejection than bully/victims. This indicates that young children regard victims as unattractive peers to get along with. This is both inconsistent [4,7], and consistent with other studies [6,35].

The bully/victim group is interesting. Bully/victims’ peer relationships followed a similar pattern to that of victims (i.e., low LM), but their SoD, PS, and CS scores more closely resembled those of bullies than those of victims or no-role children. This is consistent with Perren and Alsaker (2006) [6]. The successful application of PS and CS seems to help bullies maintain their role. From a dominance perspective, the distinction between bullies and bully/victims seems to depend on whether or not they succeed in attempting to obtain or maintain SoD. The success of such attempts seems to depend on the appropriate use of PS and CS; A previous study showed that dominant children use PS twice as much as CS [14]. Success may be related to the appropriateness or effectiveness of a certain strategy used in a certain situation, but how bullies succeed in this process is not known. Additionally, bully/victims’ PS and CS may raise their SD to certain levels (although not as high as those of bullies), but their victimization may lead them to be less liked by their peers.

The third aim of this study was to investigate the percentage of bullying role changes. The results showed how many children were involved in bullying and how many of those children changed their bullying roles over 9–10 weeks. About one-fourth of children in this study participated in bullying, both at T1 and at T2. Whether or not the similar percentages between the two time points happened coincidently remains a question. For example, it is not entirely clear whether the ratio of children involved in bullying would be similar if the measures were conducted one more time (i.e., Time 3). Measuring the changes in children’s bullying roles from a more longitudinal view, such as over an entire academic year, and studying whether 25% of children remain involved in bullying might be useful.

This study did produce some useful findings regarding role changes between T1 and T2. First, many young children repeatedly engaged in bullying behavior or were repeatedly victimized. Sixty-five percent of children involved in bullying at T1 also engaged in bullying at T2; whether or not their bullying roles changed, they maintained their involvement in certain bullying roles at least at one time point. Moreover, more than one-third of children (36.9%) involved in bullying at T1 maintained their bullying roles at T2 (i.e., same bullying role)—that was, 9.6% (127 children) of the total sample (1312 children). In other words, nearly one out of ten children engaged in bullying by maintaining the same bullying roles for two months. This reflects that many young children consistently bully others, and other children are consistently victimized.

Next, the tendency of the changes in bullying roles differed from one role to another. Bullies were more likely than victims and bully/victims to maintain their roles; 40% of bullies maintained their roles. Meanwhile, more children escaped from victim roles than children who maintained their victim roles. This indicates that victimization in early childhood is quite changeable, while the bully role is more stable than other roles—a finding that is consistent with previous research [39,40].

Third, bullies and victims rarely exchange roles; bullies can become bullies/victims or take on a no-role status, but they are unlikely to become victims; the reverse is true as well. Meanwhile, the pattern of bully/victims’ role change is partly different from those of bully, victim, and no-role children. Bullies/victims at T1 were spread into other roles more indiscriminately than bullies or victims. That is, bully/victims were more likely than other bullying roles to experience inconsistent role changes during bullying processes.

Furthermore, many bullies at T2 were either bullies or had no role at T1. Similarly, many victims at T2 were either victims or had no role at T1. Together, these role change findings indicate that early intervention is necessary to prevent repeated bullying or victimization and to discourage children from becoming newly involved in bullying or victimization.

The fourth aim of this study was to investigate whether changes in bullying roles are related to the changes in dominance and peer relationships, and the results showed many new findings. Role changes were related to changes in SoD, PS, and CS. Children that lost their bullying role had decreased SoD, PS, and CS scores, whereas, children that maintained their bully role sustained those scores. Conversely, children newly involved in bullying had increased SoD and CS scores. Few class effects were found in this process.

It seems clear that dominance and bullying role changes are closely related. Whether bullying leads to high social dominance or high social dominance elicits bullying behavior is unclear; it could be bidirectional, but previous findings have shown the former to be more likely. Reijntjes et al. [30] found that children maintaining high levels of bullying behavior had consistently high resource control levels, but the reverse was not true. Similarly, young children’s prosocial behavior predicted social dominance after two months, whereas, social dominance did not influence later prosocial behavior [59].

In this study, role changes were only sometimes related to changes in peer relationships. Children maintaining bully/victim roles had decreased LM scores, and no-role children who became bully/victims had increased LL scores.

Interestingly, children who were victims and changed to no role had increased PASS. Victimization seems to be a less attractive factor among same sex children than opposite sex children. Previous studies [36,50] showed that bullies were less liked by same sex peers, but did not show a relationship between victimization and same/opposite sex children’s likeability. Moreover, young children’s prosocial and aggressive behaviors during play might be more exposed to same sex peers, as young children had a very strong tendency to play with same sex peers. Victims may have fewer attractive resources and use fewer PS, which may impact their peer relationships.

How do victims escape victimization? This study did not examine the direction of the causality between popularity and victimization—whether same sex peers’ preferences help victims escape victimization or whether escaping victimization leads same sex peers to like former victims. Whatever the causal direction, increasing peers’ favorability towards victims can help to stop victimization. The finding that having friends can help protect against victimization supports this [70,71]. Moreover, it supports why intervention programs focus on building positive peer relationships among young children [72].

Children maintaining their bully/victim roles were exposed to risks in peer relationships; their peer relationships became worse, while they maintained their roles. In contrast, children in other roles had no changes in peer relationships if they maintained their bullying roles. This reflects that bully/victims may be a vulnerable group in peer relationships and might have more behavioral and emotional difficulties than victims [29].

Children who were not involved in bullying at both time points showed a decrease in their use of PS. The reason for this decrease is not clear, but it may be supposed that their peer relationships or their friendships were already firm, meaning they may not have needed to use PS to get along with their peers. Moreover, PS are not synonymous with prosocial behaviors. In fact, the distinction between PS and prosocial behaviors is not clear. Sometimes, using prosocial strategies is helpful and recommended for obtaining toys or social resources, but such strategies can be seen as insincere and serving-self rather than truly altruistic. Prosocial behavior is voluntarily performed for other people without consideration for its potential benefits to the performer [73]. Thus, the behavior’s effect may depend on its underlying motivation. If a study could distinguish between PS and prosocial behavior and measure their effects, the result could provide more insight regarding whether the use of prosocial strategies functions differently than engaging in prosocial behavior.

Finally, role changes were only sometimes related to changes in peer relationships, but they were more often associated with changes in dominance. At first glance, this seems counterintuitive because many studies have demonstrated the link between bullying roles and peer status. The role changes did not immediately seem to reflect the changes in peer relationships.

This might imply that peer relationships are more stable than changes in dominance or roles. In other words, peer relationships may change more slowly than roles. For example, a child not involved in bullying can become a new bully/victim and use more CS in class, but his/her friends may still like him/her for a while. They may not change their preferences quickly although their friend bullies or is victimized by other children. This may be partially supported by the findings of Reijntjes et al. [74]: Bullies in early adolescence do not experience concurrent negative consequences; rather, they experience immediate benefits, such as maintaining high perceived popularity and remain reasonably well accepted, although they are disliked to some extent by their peers.

Alternatively, bullies in early childhood can fit in well with peer groups [75]; they might affiliate with other aggressive children [75] or non-involved children with low social competence [7]. In fact, non-involved children (outsiders) are not clearly distinguished in early childhood; they may be afraid of going against bullies, or they may just concentrate on their own activities or play rather than watching the situation [7]. Therefore, the behavior of “no role” children may require further clarification. Further longitudinal research is necessary to determine the stability of changes in bullying roles in relation to dominance and peer relationships.

Lastly, there were no class effects on the prevalence of bullying, victimization, or the distribution of bullying roles in this study. Moreover, class variables did not influence dominance and peer relationships. Only a few class effects on the relationship between bullying role changes and changes in dominance were found. The effects of teachers’ carrier, sex ratio, and class size became apparent when non-involved children were newly engaged in bullying. It is not clear the reason for these results, but it can be supposed that more experienced teachers are available to manage children’s CS. This may imply that these teachers could be considered to establish prevention programs for young children’s bullying. For example, the higher the teachers’ carrier, the larger the decrease in CS over the semester. Less experienced teachers may need more support to manage children’s CS.

This study provides valuable insights into bullying in early childhood. Many findings in this study highlight the need for prevention and intervention programs as early as possible in bullying processes. First, this study shows the prevalence of bullying among young children and bullying role changes over 9-10 weeks using a large sample. The findings suggest that involvement in bullying among young children is not uncommon. One-tenth of young children remained involved in bullying over time. This implies that bullying prevention/intervention programs are necessary for children at these ages. Furthermore, the rates of role changes for bullies, victims, and bully/victims suggest that bullying interventions should occur as early as possible to prevent the creation of new victims.

Second, role changes and their connections to dominance and peer relationships can be considered strengths of this study. The findings suggest that adults should pay attention to hierarchies in young children’s classes. Many studies have examined the ratios between bullying roles, but previous studies have not reported the ratio of change in these roles. Importantly, this study provides important implications that bullying is complicatedly related to dominance and peer relationships from young ages. Bullying is a matter of relationships and power hierarchies. To prevent bullying, adults should pay attention to whether particular children occupy resources and privileges in classes, and whether their power is sustained over time or not. Some children who look kind can be dominant or bossy to other children. Teachers should be careful to avoid the biased belief that children who appear prosocial are always nice to peers.

Third, this study has bullying-related implications for kindergarten teachers. Teachers’ perceptions of bullying among children may significantly impact how they intervene and prevent it [3]. Moreover, this study showed that teachers’ reports could serve as reliable measures by producing findings that are very consistent with previous studies using peer reports [14]. Teachers need to recognize the different bullying roles and their various characteristics in terms of peer relationships and power statuses. Kindergarten teachers are likely unaware of bullying in kindergarten [76]. This study highlights the importance of teachers’ roles because they are in the best positions to observe children’s peer hierarchies.

In particular, this study showed the importance of distinguishing bully/victims from bullies. Children in the bully/victim role are more likely to change than children in other roles. Additionally, teachers often regard bully/victims as bullies, and peers rarely care about bully/victims’ victimization [41]; Perceiving bullies/victims as bullies may exacerbate bully/victims’ victimization.

This study has several limitations, particularly in terms of methodology. First, this study collected data only twice at approximately 9–10 week intervals. If the measurements had been made one more time, it would have provided more insight into the changes in bullying roles—specifically, whether or not the recorded changes/non-changes were sustained. Moreover, using both qualitative and qualitative approaches may compensate for each other; children’s bullying roles can change over time (e.g., month, semester, etc.), but they can also be influenced by certain social events, such as picnics, summer camps, or sports games. For example, a child who is capable of football may occupy a dominant and popular position after a football game. Therefore, additional research that takes a more longitudinal perspective and uses more diverse methodologies is necessary.

Second, this study mainly relied on teachers’ measurements of children’s bullying behaviors, peer relationships, and dominance, although peer relationship was measured partly by children’s reports. This was unavoidable, due to the IRB stipulations and the ethical approval of each kindergarten. Further studies need to use a more diverse methodology to investigate bullying, peer relationships, and dominance, such as peer or self-reports (if available). Alternatively, a qualitative approach would be useful for bullying in early childhood, as mentioned above.

Third, from an analysis perspective, the data of this study were analyzed using a variable-centered approach. This makes it difficult to avoid the criticism that the standards used to distinguish bullying roles were arbitrary, though this study tried to use a conservative standard (i.e., 1 SD). If a person-centered approach was applied to analyze the data, it could provide a different perspective on the relationship between bullying roles and peer relations, and dominance. For example, researchers using cluster analysis may categorize bullying roles differently.

Lastly, this study did not examine whether bullying roles, dominance, and peer relationships had different cycles of changes or not. How many times children experience role changes and how long role change cycles last during one academic year remains unknown; these changes/cycles can happen consistently and irregularly. For example, victim role changes may be shorter than those of bullies, and role changes may influence later peer statuses. If these were investigated, it would help efforts to determine the optimal period for implementing prevention/intervention initiatives and the frequency or time of interventions, and this would boost the effectiveness of prevention/intervention programs.

## 5. Conclusions

This study showed that bullying involving an imbalance of power and repeated aggressive behaviors clearly exists in early childhood; it also produced some new findings regarding the relationship between bullying roles changes and both dominance and peer relationships changes. Findings show that, while some children become involved/cease involvement in bullying over time, certain children remain involved in bullying, and their bullying roles sometimes change and sometimes do not. Intervention efforts should take a different peer relationships and social dominance levels across bullying roles into consideration. Bullies need to learn positive ways to build relationships with peers that do not involve coercive behavior based on their dominant statuses. Children in the bully/victim group need considerable attention from peers and teachers, since the flexibility of their roles and peer relationships can become worse over time. Non-involved children also need to learn how to help and intervene in bullying and how they can stand up when they witness peers treating other children unfairly. The KiVa program [77], which focuses on bystander education, can serve as a good example in this respect. Additionally, awareness of hierarchies among children and efforts to decentralize power among children can contribute to establishing democratic atmospheres in classes, thereby helping to prevent bullying, and teachers could play key roles in such efforts.

## Figures and Tables

**Table 1 ijerph-17-01734-t001:** Descriptive statistics for bullying roles, SoD, PS, and CS.

Variable	Time Point	*N*	Mean	SD
Bullying	T1	1304	2.00	0.842
	T2	1307	1.98	0.842
Victimization	T1	1305	1.81	0.697
	T2	1302	1.78	0.699
Social dominance (SoD)	T1	1301	2.75	0.872
	T2	1307	2.71	0.862
Prosocial strategy (PS)	T1	1308	2.72	0.944
	T2	1310	2.63	0.932
Coercive strategy (CS)	T1	1306	2.13	1.014
	T2	1308	2.09	0.961

**Table 2 ijerph-17-01734-t002:** Correlations among variables.

	1	2	3	4	5	6	7	8	9	10	11	12	13	14	15	16	17	18	19	20	21	22
1 Bul T1	-																					
2 Bul T2	0.76 **	-																				
3 Vic T1	0.62 **	0.51 **	-																			
4 Vic T2	0.49 **	0.62 **	0.68 **	-																		
5 SoD T1	0.56 **	0.45 **	0.17 **	0.14 **	-																	
6 SoDT2	0.43 **	0.52 **	0.14 **	0.17 **	0.71 **	-																
7 PS T1	0.51 **	0.44 **	0.23 **	0.24 **	0.60 **	0.46 **	-															
8 PS T2	0.46 **	0.56 **	0.24 **	0.31 **	0.48 **	0.56 **	0.74 **	-														
9 CS T1	0.86 **	0.72 **	0.55 **	0.45 **	0.57 **	0.45 **	0.44 **	0.40 **	-													
10 CS T2	0.72 **	0.87 **	0.47 **	0.56 **	0.46 **	0.55 **	0.39 **	0.51 **	0.77 **	-												
11 LM T1	−0.12 **	−0.10 **	−0.18 **	−0.16 **	0.15 **	0.13 **	0.07 **	0.06 *	−0.10 **	−0.09 **	-											
12 LM T2	−0.12 **	−0.10 **	−0.27 **	−0.25 **	0.15 **	0.16 **	0.05	0.04	−0.14 **	−0.12 **	0.41 **	-										
13 LL T1	0.28 **	0.25 **	0.39 **	0.34 **	0.08 **	0.09 **	−0.00	0.02	0.33 **	0.29 **	−0.24 **	−0.26 **	-									
14 LL T2	0.26 **	0.31 **	0.35 **	0.36 **	0.09 **	0.13 **	0.00	0.03	0.32 **	0.37 **	−0.19 **	−0.24 **	0.64 **	-								
15 PASS T1	−0.17 **	−0.13 **	−0.41 **	−0.32 **	0.28 **	0.24 **	0.18 **	0.13 **	−0.17 **	−0.12 **	0.38 **	0.44 **	−0.39 **	−0.33 **	-							
16 PASS T2	−0.18 **	−0.20 **	−0.41 **	−0.41 **	0.21 **	0.22 **	0.14 **	0.13 **	−0.18 **	−0.21 **	0.31 **	0.51 **	−0.36 **	−0.37 **	0.68 **	-						
17 PAOS T1	−0.24 **	−0.20 **	−0.39 **	−0.32 **	0.17 **	0.19 **	0.05	0.04	−0.24 **	−0.19 **	0.29 **	0.37 **	−0.41 **	−0.34 **	0.61 **	0.54 **	-					
18 PAOS T2	−0.21 **	−0.25 **	−0.34 **	−0.37 **	0.12 **	0.15 **	0.04	0.05	−0.22 **	−0.25 **	0.27 **	0.41 **	−0.34 **	−0.40 **	0.49 **	0.62 **	0.67 **	-				
19 Carrier	−0.02	−0.04	0.02	−0.06 *	−0.07 *	−0.05	−0.17 **	−0.18 **	0.00	−0.04	0.00	0.00	0.00	0.00	−0.08 **	−0.10 **	0.00	−0.01	-			
20 S.Rati	0.11 **	0.10 **	0.09 **	0.07 *	0.15 **	0.09 **	0.06 *	0.08 **	0.11 **	0.11 **	0.00	0.00	0.01	0.00	0.04	−0.02	0.03	0.04	0.05	-		
21 C.Size	−0.05	−0.08 **	−0.05 *	−0.09 **	−0.08 **	−0.08 **	0.04	−0.04	−0.08 **	−0.08 **	0.00	0.00	0.00	0.00	−0.06 *	−0.08 **	−0.05	−0.08 **	−0.08 **	0.02	-	
22 C.Age	−0.11 **	−0.12 **	−0.14 **	−0.17 **	−0.07 **	−0.03	−0.07 **	−0.07 *	−0.10 **	−0.09 **	0.00	0.01	0.00	−0.01	0.03	-0.02	0.05	0.03	0.02	0.11 **	0.47 **	-
23 Ask.Like	−0.02	−0.01	−0.01	−0.02	−0.03	0.04	−0.02	0.02	−0.05	−0.01	0.00	0.00	0.00	0.00	−0.02	0.06 *	−0.01	−0.02	0.16 **	0.01	−0.05	0.01

Bul: Bullying; Vict: Victimization; SoD: Social Dominance; PS: Prosocial Strategies; Coercive Strategies; LM: Like-Most: LL:Like Least; PASS:Peer acceptance among Same Sex; PAOS:Peer acceptance among Opposite Sex; Carrier: Teachers’ Carrier; S.Rati; Sex ratio in a class; C. Size: Class Size; C.Age: Class Age; Ask.Like.: Ask likeability directly. * *p* < 0.05, ** *p* < 0.01.

**Table 3 ijerph-17-01734-t003:** Number (percentage) of children in each bullying role.

Bullying Roles	Number of Children (%)	
	Time 1	Time 2 (%)
Bully	130 (9.7%)	132 (10%)
Victim	102 (7.8%)	123 (9.3%)
Bully/victim	112 (8.6%)	84 (6.4%)
No role	968 (74%)	973 (75%)
Total	1312 (100%)	1312 (100%)

**Table 4 ijerph-17-01734-t004:** Means (standard deviations) of SoD, PS, CS, LM, LL, PASS, and PAOS by bullying roles.

		Bully	Victim	Bully/Victim	No Role	F (*df*)	Partial η^2^
Social dominance (SoD)	T1	3.60 (.800) ^a^	2.65 (0.626) ^b^	3.19 (0.667) ^c^	2.60 (0.846) ^b^	67.429 (3, 1288) ***	0.137
	T2	3.52 (0.855) ^a^	2.68 (0.741) ^b^	3.10 (0.704) ^c^	2.58 (0.820) ^b^	59.569 (3, 1297) ***	0.121
Prosocial Strategies (PS)	T1	3.35 (0.886) ^a^	2.60 (0.840) ^b^	3.10 (0.808) ^a^	2.60 (0.929) ^b^	32.819 (3, 1288) ***	0.070
	T2	3.41 (0.831) ^a^	2.67 (0.882) ^b,c^	2.94 (0.794) ^b^	2.49 (0.905) ^c^	45.117 (3, 1297) ***	0.094
Coercive Strategies (CS)	T1	3.34 (0.873) ^a^	2.55 (1.031) ^b^	3.23 (0.932) ^a^	1.80 (0.793) ^c^	207.196 (3, 1288) ***	0.328
	T2	3.20 (0.877) ^a^	2.41 (0.935) ^b^	3.21 (0.909) ^a^	1.80 (0.767) ^c^	186.170 (3, 1297) ***	0.301
Like most (LM)	T1	−0.05 (0.901) ^a,b^	−0.35 (0.965) ^b^	−0.35 (0.844) ^b^	0.09 (0.984) ^a^	11.91 (3, 1271) ***	0.027
	T2	0.14 (1.078) ^a^	−0.60 (0.717) ^b^	−0.53 (0.738) ^b^	0.10 (0.967) ^a^	27.581 (3, 1267) ***	0.065
Like least (LL)	T1	0.35 (1.055) ^a^	1.00 (1.601) ^b^	0.945 (1.400) ^b^	−0.26 (0.600) ^c^	124.539 (3, 1271) ***	0.227
	T2	0.327 (1.066) ^a^	0.74 (1.475) ^b^	1.29 (1.469) ^c^	−0.25 (0.622)^d^	122.023 (3, 1267) ***	0.224
Peer acceptance among same sex (PASS)	T1	3.49 (0.846) ^a^	2.70 (0.782) ^b^	2.80 (0.794) ^b^	3.60 (0.787) ^a^	66.476 (3, 1271) ***	0.136
	T2	3.50 (0.937) ^a^	2.78 (0.822) ^b^	2.73 (0.887) ^b^	3.60 (0.789) ^a^	59.004 (3, 1267) ***	0.123
Peer acceptance among opposite sex (PAOS)	T1	2.95 (0.813) ^a^	2.51 (0.730) ^b^	2.42 (0.761) ^b^	3.24 (0.758)	60.786 (3, 1271) ***	0.125
	T2	2.97 (0.943) ^a^	2.48 (0.888) ^b^	2.37 (0.798) ^b^	3.21 (0.770) ^c^	52.598 (3, 1267) ***	0.111

The different subscripts (^a,b,c,d^) indicate significant differences using *Scheffe* (*p* < 0.001), except for one case; the difference between bully and no role on PAOS at T2 (*p* < 0.05). *** *p* < 0.001. T1, Time 1; T2, Time 2.

**Table 5 ijerph-17-01734-t005:** Bullying role changes from T1 to T2 and SD, PS, and CS means (standard deviations) at T1 and T2 by each role change. The cells indicated with bold style show significant mean differences between T1 and T2. Significance level is shown in the cell at T2: * *p* < 0.05, ** *p* < 0.01, *** *p* < 0.001.

Role at T1 (N)	Role at T2	*N* (Percentage of Role at T1)	SoD Mean (SD)		PS Mean (SD)		CS Mean (SD)	
			T1	T2	T1	T2	T1	T2
Bully (130)	Bully	54 (41.5)	3.91 (0.742)	3.84 (0.724)	3.59 (0.820)	3.63 (0.781)	3.55 (0.720)	3.42 (0.825)
	Victim	8 (6.2)	3.44 (0.695)	2.81 (0.828)	3.10 (1.280)	2.74 (0.762)	3.38 (0.733)	3.00 (0.777)
	Bully/victim	19 (14.6)	3.29 (0.676)	3.12 (0.643)	3.04 (0.795)	2.84 (0.702)	3.30 (0.800)	2.92 (0.748) **
	No role	49 (37.6)	3.39 (0.806)	2.86 (0.705) ***	3.19 (0.889)	2.83 (0.802) **	3.10 (1.006)	2.51 (0.947) ***
Victim (102)	Bully	5 (4.9)	3.23 (0.560)	3.80 (0.617)	3.03 (0.298)	3.33 (0.773)	2.75 (0.776)	3.21 (0.886)
	Victim	40 (39.2)	2.55 (0.520)	2.50 (0.612)	2.48 (0.840)	2.52 (0.810)	2.60 (1.00)	2.45 (0.867)
	Bully/victim	12 (11.7)	2.65 (0.539)	3.25 (0.452) *	2.65 (0.866)	2.78 (0.649)	3.00 (1.0225)	3.60 (1.021) *
	No role	45 (44.1)	2.66 (0.705)	2.43 (0.724) **	2.65 (0.867)	2.59 (0.774)	2.33 (1.074)	2.07 (0.703) *
Bully/victim (112)	Bully	31 (27.7)	3.36 (0.656)	3.21 (0.722)	3.39 (0.684)	2.98 (0.701) **	3.22 (0.964)	2.80 (0.929) **
	Victim	21 (18.6)	3.30 (0.557)	3.16 (0.845)	3.024 (0.725)	2.92 (0.895)	3.08 (0.921)	2.80 (0.924)
	Bully/victim	33 (29.5)	3.15 (0.795)	2.96 (0.789) *	3.14 (0.766)	3.06 (0.825)	3.50 (0.940)	3.35 (0.891)
	No role	27 (24.1)	3.01 (0.556)	2.77 (0.845)	2.80 (0.888)	2.70 (0.877)	3.07 (0.823)	2.30 (2.296) ***
No role (968)	Bully	42 (4.3)	3.11 (0.946)	3.34 (0.981) *	3.22 (0.926)	3.43 (0.890)	2.67 (0.800)	3.19 (0.851) **
	Victim	54 (5.5)	2.63 (0.778)	2.63 (0.729)	2.66 (0.962)	2.65 (0.955)	1.97 (0.844)	2.15 (0.938)
	Bully/victim	20 (2.1)	2.74 (0.808)	3.23 (0.734)	2.73 (0.861)	2.92 (0.920)	2.43 (0.740)	3.02 (0.940) *
	No role	852 (88.0)	2.57 (0.840)	2.56 (0.827)	2.56 (0.918)	2.46 (0.910) ***	1.73 (0.751)	1.72 (0.731)

**Table 6 ijerph-17-01734-t006:** Hierarchical regression for predicting SoD, PS, and CS differences, including class variables (Group 1: Bully group).

	SoD Difference (SoD T1-SoD T2)				PS Difference (PS T1-PS T2)				CS Difference (PS T1-PS T2)			
IVs	Model 1		Model 2		Model 1		Model 2		Model 1		Model 2	
	*B*	*p*	*B*	*p*	*B*	*p*	*B*	*p*	*B*	*p*	*B*	*p*
Role change	0.276	0.003	0.260	0.005	0.256	0.006	0.234	0.012	0.273	0.003	0.270	0.005
Carrier			−0.162	0.088			0.074	0.434			−0.049	0.609
Sex ratio			0.163	0.088			0.041	0.665			0.038	0.692
C. Size			0.090	0.355			0.216	0.030			−0.009	0.931
C. Age			−0.116	0.249			−0.085	0.397			−0.066	0.519
Model *F*	*F*(1,111) = 9.173, *p* = 0.003		*F*(5,107) = 3.108, *p* = 0.012		*F*(1,111) = 7.811, *p* = 0.006		*F*(5,107) = 2.658, *p* = 0.026		*F*(1,112) = 9.024, *p* = 0.003		*F*(5,108) = 1.898, *p* = 0.101	
R^2^ (Adj.R^2^)	0.276 (0.068)		0.356 (0.086)		0.256 (0.057)		0.332 (0.069)		0.273 (0.066)		0.284 (0.038)	

Note: The differences between T1 and T2 in SoD, PS, CS were calculated by subtracting means of T2 from those of T1. Carrier: Teachers’ Carrier; C. Size: Class Size; C. Age: Class Age.

**Table 7 ijerph-17-01734-t007:** Hierarchical regression for predicting CS differences, including class variables (Group 3: Bully/victim group).

		CS Difference (CS T1-CS T2)			
IVs	Model 1			Model 2	
	*B*		*p*	*B*	*p*
Role change	0.224		0.031	0.219	0.033
Carrier				−0.205	0.051
Sex ratio				0.015	0.886
C. Size				−0.100	0.404
C. Age				−0.078	0.519
Model F	*F*(1,91) = 4.816, *p*= 0.031			*F*(5,87) = 2.353, *p* = 0.047	
R^2^ (Adj.R^2^)	0.224 (0.040)			0.345 (0.069)	

Note: CS: Coercive Strategies; Carrier: Teachers’ Carrier; C. Size: Class Size; C. Age: Class Age.

**Table 8 ijerph-17-01734-t008:** Hierarchical regression for predicting SoD, PS, CS, and LL differences, including class variables, and a method for investigating LM and LL (Group 4: No-role group).

	SoD Difference (SoD T1-SoD T2)				PS Difference (PS T1-PS T2)				CS Difference (CS T1-CS T2)				LL Difference (LL T1-LL T2)			
IVs	Model 1		Model 2		Model 1		Model 2		Model 1		Model 2		Model 1		Model 2	
	*B*	*p*	*B*	*p*	*B*	*p*	*B*	*p*	*B*	*p*	*B*	*p*	*B*	*p*	*B*	*p*
Role change	−0.095	0.006	−0.101	0.002	−0.119	0.001	−0.118	0.001	−0.168	0.000	−0.166	0.000	−0.093	0.008	−0.095	0.008
Carrier			−0.061	0.075			0.015	0.003			0.076	0.027			−0.024	0.503
Sex ratio			0.115	0.001			−0.023	0.502			−0.002	0.949			0.014	0.697
C. Size			0.022	0.569			0.128	0.001			0.030	0.435			0.015	0.700
C. Age			−0.102	0.008			−0.064	0.696			−0.057	0.133			−0.006	0.882
Asking L.															−0.014	0.691
Model F	*F*(1,837) = 7.574, *p* = 0.006		*F*(5,833) = 5.601, *p* = 0.000		*F*(1,844) = 12.120, *p* = 0.001		*F*(5,840) = 4.876, *p* = 0.000		*F*(1,840) = 24.515, *p* = 0.000		*F*(5,836) = 6.292, *p* = 0.000		*F*(1,794) = 6.93, *p* = 0.008		*F*(6.789) = 1.337, *p* = 0.238	
R^2^(Adj.R^2^)	0.095 (0.008)		0.180 (0.027)		0.119 (0.013)		0.168 (0.022)		0.168 (0.027)		0.190 (0.031)		0.093 (0.007)		0.100 (0.003)	

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
