# Peer review of "Kindergarten Teachers’ Perspectives on Young Children’s Bullying Roles in Relation to Dominance and Peer Relationships: A Short-Term Longitudinal Approach in South Korea"

_ijerph, 2020, doi:10.3390/ijerph17051734_

Round 1

Reviewer 1 Report

Thank you for addressing the comments. 

Author Response

Thank you very much for your time for reviewing the manuscript. Your comments were very helpful for improving the manuscript.

Reviewer 2 Report

I have no relevant comments about the investigation. I think it is a good contribution to this field of study. 

Author Response

Thank you very much for your time for reviewing the manuscript.

Reviewer 3 Report

Thank you for the opportunity to review this very interesting manuscript entitled "Young children’s bullying roles in relation to dominance and peer relationships: A short-term longitudinal approach in South Korea".  The authors have done an effort to analyze the problem of bullying among children (early age 3-5) from the perspective of the teachers of public kindergartens. However, I consider that the authors should change some aspects of the manuscript to publish it in this journal. My decision is “Reconsider after major revision”.

Title:

I consider that the title “Young children’s bullying roles in relation to dominance and peer relationships: A short-term longitudinal approach in South Korea” does not show the meaning of the manuscript because this manuscript analyze the bullying problem from the perspective of the teachers. This aspect should be present in the title.

Abstract:

The abstract could be improved in some aspects. There are several studies about peer-relationships and bullying, the authors should change this sentence “Few studies have investigated young children’s bullying roles in relation to dominance” and change it. For example, the author could show the importance to analyze these variables on the bullying. According the instructions for authors “the abstract should be a total of about 200 words maximum”. The abstract of this manuscript has 269 words, the authors should reduce the abstract.

Introduction:

The introduction should be more focused on the aggressiveness of children of early ages. This change should be incorporate in the beginning of the section 1.1. Bullying, dominance, and peer relations.

Also, the authors should explain the aims and hypothesis of this manuscript clearer and more concise. In this part of the manuscript the authors should not comment previous results in others studies or conclusions.

Method:

In the “Participants” section the authors should describe the exactly reason to administrate two times the instruments to the teachers in a few time (two months).

Also, the author should describe better the procedure. How many classrooms were selected in each school? How long did the data collection process take overall? How many time did the participants employ to fill the survey?

Results:

The authors should order the results in the correct order. The results do not correspond with the order of the aims or “statistical analysis” section. The Table 1 is not commented. The authors comment the regression analysis in the beginning and this is not correct. Also, in the section “3.1. Relationship between bullying roles and social dominance and prosocial and coercive strategies” the authors comment the MANOVA, and they did not comment the correlations among the variables. I recommend to the authors check the correct order of the results and the tables.

Discussion and conclusions:

The description of the implications and suggestions should be in the discussion section.

References:

The style of the references is correct according the instructions of this journal: references must be numbered in order of appearance in the text (including table captions and figure legends) and listed individually at the end of the manuscript.

I recommend to check again this section after the changes in the manuscript.

Author Response

Thank you very much for your time for reviewing the manuscript. I attached the cover letter which indicates the reivisions made.

Round 2

Reviewer 3 Report

Thank you for the opportunity to review this very interesting manuscript entitled " Kindergarten teachers’ perspectives on young 3 children’s bullying roles in relation to dominance and 4 peer relationships: A short-term longitudinal approach 5 in South Korea". The authors have done an effort to check and modify according to my suggests. Taking account this, like a reviewer of this manuscript now I consider that this manuscript could be published.

This manuscript is a resubmission of an earlier submission. The following is a list of the peer review reports and author responses from that submission.

Round 1

Reviewer 1 Report

This is a substantial stufy. It essentially consists of a series of meadures administered twice.  I I made the fattached comments on p1 which deal with easily remediable style issues. Also I think there is a raft of work by David Farrington and colleagues which may be useful in clarifying the argument made here.

When I read on I encountered issues which I think need addressing. These are as follows.

Each teacher rated the same children on both occasions thus precluding assessment of reliability..

2. The same teachers assessed the same children on both occasions, meaning that the first ratings may have sensitized to the second, hence making the change measure problematic.

3. I would like to see between teacher comparison of the association between dominance and bullying role. There seems to me to be a danger of these measures being conflated. Do children bully because they are dominent or does teacher observation of bullying lead to the inference of dominance.

With hard won data, I would l compare teacher ratings. If there is no selection of children, expected bullying rates should be similar across teach in more detailers. Between teacher variability (eg by teacher age) would be interesting and practically useful and useful to look at. I am in no way an expert but there were a number of warning signs that the analauses needs looking at in depth.

I found the account of results really difficult to follow and think there should be a statistical second judgement of the aapproach taken. For example, was a Bonferroni correction xonsidered? Was the Scheffe test chosen on the basis of its extreme position on per comparison error rate?

Associations between bullying roles, dominance and 3 peer relationships among young children in South 4 Korea; A short-term longitudinal approach

Abstract

L9 3) the ratio of bullying role change/maintenance over time. It will no doubt be clearer in the body of the paper, but the phrasing is unclear. Is it perhaps role change/n   ??

P1 l22 who finds self defence difficult.

P1 l45 Is this relationship inverse?

P1 l52 aggressive submission?

Reviewer 2 Report

This study addresses an interesting and important topic on the relationship between bullying roles, dominance, and peer relationships in South Korea. This work offers a novel contribution using primary data collected from a large number of students. However, as I detail below there are several shortcomings of the paper that lead me to question the robustness of the findings and overall contribution. 

1. On line 15 the authors say “data was collected.” Authors should note that data are plural. All phrases should be changed to say “data are” “data were”

2. Several sentences should be reworded for clarity in the front end of the paper. For instance, lines 31-33 are confusing. Additionally, the passage (lines 36-49) should be re-worked. Regarding the latter, the authors describe various perspectives and then after doing so, provide the name of the perspective. Instead, the author should lead with the name of the perspective. For example, move up Hawley’s Resource Control Theory earlier in the passage and then describe what it is and how it fits with the study. 

3. Is line 50 (Bullying, dominance, and peer relationships) supposed to be a new section? 

4. There is a typo on line 51. There are also several typos throughout and the manuscript should be carefully edited. 

5. Can the author please elaborate on lines 161-162 about what is meant by “most of the teachers did not have any problems completing the questionnaire” What percentage is most? What happened to teachers that did have issues? Were there incomplete questionnaires? If so, how many and how were those handled? 

6. Please clarify the discussion on lines 168-170, “teachers were allowed to ask their children directly about peer relationships when possible. Some teachers did not agree with this approach and preferred to respond to the questions themselves.” How many teachers interreacted with students and how many did not. Is there a chance that directly interacting with students would bias the responses? Perhaps the author could show a sensitivity analysis about whether response patterns differed between teachers who did and did not interact with students. 

7. The time point between the two surveys was 9.5 weeks on average. I am concerned about the amount of change that could actually take place during this time period. Could the author please justify why this is a reasonable amount of time to observe change in behavior for this age range of students. 

8. In the United States, children are not in school for much of December and January because of religious holidays in December and New Years in January. I am not sure if this is the case in South Korea, but if so, that would mean toward the end of the survey students were out of school for a 2-3 week period during which time they were mostly with family or in daycare. This might influence the results of bullying and social dynamics as children are returning to a school setting following an extended period of time away. Could the author please clarify whether this is the case in South Korea. If so, then I am concerned the results are at least partially influenced by this recent extended period of time out of school as it would likely impact bullying/peer dynamics. 

9. My biggest concerns deal with the methodological approach selected by the authors. First, the study analyzes data on over 1,300 students nested within 63 teachers and choses a t-test to detect significant differences between the two time points. However, this appears to be an inappropriate strategy given the nested nature of the data as all the characteristics being studied here (bullying, victimization, social dominance, etc) are likely influenced by classroom dynamics and students are nested within classrooms. Therefore, the study should account for the clustering of students within classrooms/schools. I would recommend doing this with either a regression analysis that clusters on the level 2 variable (teacher/classroom/school) or to employ a multilevel modeling strategy. Without accounting for this, the standard errors are likely biased. 

10. Building off the above comment, I am not sure why the author did not use a multivariate regression. The authors clearly devoted time to collect a rich set of data, but then employs a t-test. This cannot rule out the possibility of spuriousness. Does the author have any other variables about the students or teachers that could be controlled for? If so, I would highly suggest that the author re-work the analysis to use a multivariate regression that includes a set of control variables and clusters the standard errors on teacher/classroom. 

11. I suggest the author provide a little more background about schooling/bullying in South Korea. I think it is very interesting to study behavior in this context. However, many readers might not have great familiarity with South Korea schooling system and bullying culture, and therefore would benefit from some additional detail. 

Reviewer 3 Report

Dear. Author

Thank you for your endeavor to study bullying phenomena among young Korean children. This study collected huge size of the sample and reflected the third parties perspectives (teachers) on bullying. However, it was little confused because of following reasons. It would be great if an author tries to modify this manuscript based on the following suggestions.

1. In this manuscript, the authors used both the concepts of likeability and popularity. However, according to the measurement, it looks very similar. Usually, these concepts are different based on previous research. To be more clearness, please kindly ask you to provide "clear and exact definitions" for these two different concepts in the introduction. 

2. If you translated all measurement what you used in this study, did you confirm the validity 

       and internal consistency? Also, I am curious whether you applied double translated

       (translated- back-translated) process or not. Please describe it more specifically.

3. Through the manuscript, authors used "association" or "relation" when describing the results.

       Since this study analyzed data with only "t-test" and "ANOVA", it is hard to use those kinds of

       words. Please be careful to use these expression when describe your results. it should be

       accordance in your analysis. 

 4. It would be very great if authors integrate words expression. It was very confusing

       because the authors applied various words to express the same concepts.

 5. I am also curious why authors considered "sex" when they collect data sets for

      measuring popularity. If the authors thoughts the roles of sex are important to measure

      popularity, it should be explained in the introduction.

 6. In the introduction, except bullying, it was hard to find clear definitions of what authors used in

     the study. Sometimes authors considered subtypes of main variables, however, it didn't

     be explained well in the introduction.

  7. There are minor things that should be corrected. (see line 329, 572...; expressions for

      statistics outcome ... should be italic. Now it is inconsistently applied.)

  8. If early interventions are important, please describe why it is important to children's

      development. For example, if negative outcomes of bullying are described in the manuscript,

      it will have more rational.

  9. In this study, the authors considered "bully" "victim" and "bully-victim" as bullying roles.

     However, diverse studies considered 6 participants roles(bully, victim, defender, assistant,

     a reinforcer and outsider) for bullying study. It will be great if you describe why you decided to

     apply 3 types of roles for categorizing in this study.

Thanks.